# Global Trends and Emerging Frontiers in Smoking and Smokeless Tobacco Research: A Bibliometric Analysis over the Past Decade

**DOI:** 10.3390/healthcare13111224

**Published:** 2025-05-23

**Authors:** Saumya Richa, Sumaila Praveen, Ahmed A. Albariqi, Shahabe Saquib Abullais, Syed Esam Mahmood, Awad Alsamghan, Rishi Kumar Bharti, Ghadah Khaled Bahamdan

**Affiliations:** 1Department of Psychology, Lovely Professional University, Phagwara 144402, Punjab, India; saumyaricha@gmail.com; 2Department of Anesthesia and Operations, College of Applied Medical Sciences, King Khalid University, Muhayil Asir 61421, Saudi Arabia; srahman@kku.edu.sa; 3Department of Periodontics and Community Dental Sciences, College of Dentistry, King Khalid University, Abha 62529, Saudi Arabia; aaalbarqi@kku.edu.sa (A.A.A.); sshahabe@kku.edu.sa (S.S.A.); ghbahamdan@kku.edu.sa (G.K.B.); 4Department of Family & Community Medicine, College of Medicine, King Khalid University, Abha 62529, Saudi Arabia; asoman@kku.edu.sa (A.A.); brishi@kku.edu.sa (R.K.B.)

**Keywords:** tobacco, oral health, smoking, smokeless tobacco, risk factors, bibliometrics, e-cigarettes

## Abstract

Tobacco use remains a critical global health issue, with extensive research focusing on its impact on public health, particularly its strong association with oral cavity cancer. It is a leading cause of preventable disease and death worldwide, affecting millions each year. Despite increased awareness and regulatory measures, tobacco continues to pose significant challenges, prompting ongoing investigations into its health effects and related behaviors. Objective: This study aims to conduct a bibliometric analysis of smoking and smokeless tobacco research from 2014 to 2024, focusing on identifying key research trends, influential contributors, emerging topics, and collaborative networks on a global scale. Methods: A dataset of 2694 research papers from PubMed was analyzed using bibliometric tools. Keyword co-occurrence, authorship patterns, and institutional collaborations were mapped to reveal dominant themes and trends. Additionally, country-specific publications were examined to assess geographical contributions and emerging research frontiers. Results: The analysis indicates a 7.3% annual increase in publications, with a peak in 2021 likely influenced by COVID-19. Research topics have shifted from traditional tobacco-related health impacts, such as lung cancer and cardiovascular diseases, to newer areas like e-cigarettes and social determinants of health. Strong international collaborations were noted, with the U.S., China, and Europe as dominant contributors. Emerging research frontiers include electronic nicotine delivery systems and strategies aimed at controlling tobacco-related health risks. Conclusion: This bibliometric study highlights significant growth in tobacco-related research over the past decade. Evolving trends reflect a shift toward newer tobacco products and public health challenges. These findings provide valuable insights for shaping future research agendas and informing global tobacco control policies.

## 1. Introduction

Tobacco, derived from Nicotiana tabacum, is a significant plant within the Solanaceae family, widely known for its nicotine content, which makes it toxic to humans. Nicotine, the primary addictive substance in tobacco, is responsible for both physical and psychological dependence. Tobacco-related illnesses claim millions of lives each year, with over 8 million deaths in 2019 alone. Despite declining usage, tobacco continues to cause significant health problems. The WHO Framework Convention on Tobacco Control (WHO FCTC) was adopted in 2003, recommending evidence-based measures for reducing tobacco demand [1]. Historically, tobacco use can be traced back to 1 B.C.E. among Native Americans [2]. Smokeless tobacco, like chewing tobacco and snuff, exposes users to over 25 known carcinogens, with tobacco-specific nitrosamines (TSNAs) being the most harmful. TSNA levels vary by product, with higher levels increasing cancer risk [3,4]. Both the International Classification of Diseases, 10th Revision (ICD-10), and the Diagnostic and Statistical Manual of Mental Disorders, 5th Edition (DSM-5), classifies tobacco use disorder as a condition influenced by genetic predisposition, environmental factors, stress, peer pressure, nicotine addiction, and co-occurring mental health conditions. For example, individuals with mental illness are twice as likely to smoke compared to the general population [5], underscoring the complex relationship between tobacco use and mental health as nicotine’s effects on mood and stress continue to be researched [6,7,8,9,10]. In India, the Global Adult Tobacco Survey (GATS) from 2009–10 and 2016–17 revealed that the country is the second-largest producer and consumer of tobacco globally. Smokeless tobacco products such as khaini (surti), gul, gutkha, betel quid with areca nut, and zarda, as well as smoking forms like cigarettes, bidis, hookahs, and pipes, are widespread. In particular, smokeless tobacco is associated with oral lesions and cancer [11,12], while public spitting of tobacco increases the risk of communicable diseases like tuberculosis. Nicotiana tabacum remains the principal commercial crop among India’s 70 varieties of tobacco plants.

Additionally, the rise of electronic cigarettes (e-cigarettes) as an alternative to traditional smoking has raised public health concerns. Research indicates that e-cigarette use is associated with an increased likelihood of transitioning to cigarette smoking, particularly among non-smoking youth [13]. Despite being marketed as safer alternatives, these products continue to contribute to nicotine dependence and health risks.

Tobacco plants also absorb toxic heavy metals like lead (Pb), cadmium (Cd), and zinc (Zn), which further complicate the health risks associated with their use [14,15]. Moreover, the harmful effects of tobacco on reproductive health are well documented. Women who smoke during pregnancy face a higher risk of complications, including infertility, miscarriage, and congenital disabilities, while men are at greater risk for erectile dysfunction and reduced sperm quality [16].

Oral cavity cancer is a common malignancy worldwide [17], affecting both developing and developed nations [18]. Approximately 405,000 new cases are reported annually, with the highest incidence in Sri Lanka, India, Pakistan, Bangladesh, Hungary, and France [19]. The oral cavity includes the lips (front), cheeks (sides), floor of the mouth (bottom), palate (top), and oropharynx (back), which extends from the hard and soft palates to the area behind the tongue’s circumvallate papillae. These structures are supported by the maxillae and mandible [20]. Tobacco use is a major cause of oral cancer, containing carcinogens such as polycyclic hydrocarbons and nitrosamines. Practices like chewing betel nuts and tobacco are widespread in Asian populations. The risk of squamous cell carcinoma of the oral cavity (SCCOC) increases with the amount of tobacco consumed (pack-years). The risk of SCCOC increases with cumulative exposure (measured in pack-years), though cessation significantly reduces this risk. For example, individuals who abstain from tobacco use for 1–9 years have a 30% lower risk (OR = 0.7, 95% CI: 0.5–1.1), while those who quit for more than 9 years see a 50% reduction in risk (OR = 0.5, 95% CI: 0.3–0.7) [21]. Previous bibliometric studies have primarily addressed isolated aspects of tobacco use, such as smoking during pregnancy or the effects of secondhand smoke [22,23]. However, there remains a lack of comprehensive bibliometric research that encompasses both smoking and smokeless tobacco, particularly in the Indian context. This gap is especially significant given India’s unique consumption patterns, diverse product types, and substantial public health burden.

Moreover, widespread misconceptions about the relative safety of certain tobacco products—especially smokeless tobacco and e-cigarettes—continue to contribute to their use, particularly among youth. These misconceptions, combined with targeted marketing and limited public awareness, have hindered effective prevention efforts.

This study seeks to address these gaps by conducting a comprehensive bibliometric analysis of global and Indian tobacco research over the past decade (2014–2024). It highlights key publication trends, risk factors, influential contributors, and policy-relevant insights. Particular attention is paid to addressing misinformation and its impact on vulnerable populations, including adolescents and pregnant women. This paper aims to map the research landscape of tobacco-related studies through bibliometric techniques, focusing on both smoking and smokeless tobacco trends globally and in India.

## 2. Methods

Formulation of Research Questions

Research questions serve as the foundation for a structured and focused study. In this case, the research aims to analyze trends, contributions, and emerging themes in smoking and smokeless tobacco studies over the past decade (2014–2024). Table 1 presents the research questions along with their significance in guiding the study.

### 2.1. Data Sources and Publications

This study is a retrospective bibliometric analysis reviewing articles published in PubMed, chosen for its accessibility and extensive healthcare database. PubMed, a vital resource of the U.S. National Library of Medicine, offers access to high-quality, peer-reviewed studies on tobacco use, smoking cessation, and public health interventions. Additionally, it utilizes the Medical Subject Headings (MeSH) indexing system to enhance search accuracy by systematically organizing research topics [24]. Since June 1997, PubMed has offered free and unlimited access to the internet, evolving beyond a public interface for MEDLINE citations to include full-text links. PubMed has been widely used in bibliometric studies analyzing tobacco-related research trends. For example, Zhang et al. (2022) used PubMed data to assess research hotspots and emerging trends in electronic cigarettes [25]. Similarly, other studies have relied on PubMed to analyze global smoking patterns, tobacco control measures, and associated health risks. Given PubMed’s comprehensive coverage of biomedical and public health research, it was selected as the primary database for this study [26]. Although Scopus and Web of Science offer extensive citation metrics, PubMed was specifically chosen for its strong focus on biomedical and healthcare research, ensuring more relevant results for tobacco-related studies. Additionally, PubMed provides free access to its data, making it a practical and widely used source for bibliometric research in public health [27,28]. Data were downloaded from PubMed on 8 September 2024. The study covers research published between 2014 and 2024, analyzing 2694 papers globally. A query was applied to the PubMed core database, focusing on titles, abstracts, and keywords. Only English-language publications were included, and the analysis considered specific document types, such as original research articles, review articles, and meta-analyses. Each entry was carefully examined to remove irrelevant information, and data on titles, abstracts, authors, countries/regions, affiliations, document types, journals, and other key metrics were extracted.

### 2.2. Data Analysis

Bibliometric analysis is a statistical method used to quantitatively examine literature and provide an overview of research areas [29]. For this analysis, R, a free, open-source tool with extensive packages for psychological analyses, was used. A large community supports R and offers reproducibility features, like R Markdown, which enhance collaboration and open science [30]. In the analysis, terms such as keywords, countries, institutions, and authors are represented by circles. The distance between circles indicates the strength of the relationships between terms. Clusters are color-coded, with circle size reflecting term frequency and line thickness representing the strength of the connection [31].

## 3. Results

The results are based on primary information about data taken from R Biblioshiny analysis on 14 September 2024. Data were taken from 2014 to 2024; the dataset includes 2694 documents from 1147 journals, books, and other sources, with a 7.3% annual growth rate. The average age of documents is 5.08 years, and the analysis includes 8445 Keywords Plus (I.D.). The most frequently occurring words are “humans” (2694), “female” (2360), and “male” (1672), followed by terms like “risk factors” (875), “adult” (678), and “pregnancy” (652). Specific health-related terms such as “middle-aged” (582), “adolescent” (302), and “smoking/adverse effects” (252) further highlight the scope of the research (Figure 1).

This figure summarizes key bibliometric indicators, including the time span (2014–2024), document count (2694), source types (1147), keyword frequency (8445), annual growth rate (7.3%), and document age (5.08 years). The rising trend in tobacco-related research reflects the global increase in scientific publications, particularly in health-related disciplines. Tracking publication trends across journals, authors, and countries allows public health institutions to identify knowledge gaps, allocate funding more efficiently, and design informed policies [32,33]. Bibliometric analysis is widely used to assess research activities and identify emerging trends in specific fields [34].

For dataset selection, the study used the following keywords in the PubMed database: “tobacco”, “smoking”, “electronic cigarettes”, “smokeless tobacco”, “nicotine addiction”, “tobacco cessation”, “cigarette smoking”, “tobacco control”. Similar keyword-based searches have been used in previous bibliometric studies on tobacco research. For instance, a bibliometric study was conducted on electronic cigarettes using the key search terms “e-cigarette”, “vaping”, and “ENDS”, while another study analyzed tobacco policy research using the key terms “tobacco control” and “smoking regulation” [26,35]. Notably, the publication peak in 2021 may be attributed to heightened research interest during the COVID-19 pandemic, which intensified focus on smoking-related respiratory health risks [34]. This trend illustrates how global health crises can shape academic output in specific fields. Overall, while the growth in tobacco-related publications mirrors global scientific trends, it is also driven by context-specific events and policy changes.

Overview

RQ. 1. How have research trends and focus areas in smoking and smokeless tobacco studies evolved over the past 10 years (2014–2024)?

This bibliometric analysis evaluates the scientific output on smoking and smokeless tobacco research from 2014 to 2024. The study addresses critical research questions regarding scientific production, leading countries, thematic trends, collaboration networks, and critical journals. Through a review of 2694 research papers sourced from PubMed, the analysis provides insights into global trends, collaboration networks, and emerging research themes in this domain.

Annual Scientific Production

Bibliometric analysis studies scientific literature by evaluating publications in a specific field or academic journal over time. It utilizes numerical and statistical methods to assess the annual production of research articles, as referenced in [34]. Figure 2 showcases the yearly trends in tobacco-related research from 2014 to 2024. The data indicate substantial fluctuations in research output across the years. From 2014 to 2016, there was a dramatic rise in publications, escalating from 84 articles in 2014 to over 330 in 2016, signaling an increasing interest in tobacco-related health issues. However, after 2017, the number of publications started to decrease, hitting a low of 225 articles in 2019. This reduction may reflect changes in research funding priorities or shifts in public health focus areas. A resurgence was noted in 2021, coinciding with the COVID-19 pandemic, which rekindled discussions on respiratory health, smoking-related comorbidities, and the effects of tobacco use during the pandemic. Between 2021 and 2023, the number of publications remained fairly stable, fluctuating between 250 and 270 articles per year. In contrast, 2024 saw a significant decline in publications, potentially indicating a shift in research focus or a decrease in funding availability. Thematic analysis reveals that key research topics over the past decade have consistently included lung cancer, cardiovascular diseases, and the health implications of both smoking and smokeless tobacco. Importantly, after 2017, there has been an increasing focus on electronic nicotine delivery systems (e-cigarettes) and the social determinants of tobacco use, highlighting shifting research priorities and ongoing regulatory debates.

Sources

RQ. 2. Which authors, institutions, and countries have been most influential in smoking and smokeless tobacco research over the last 10 years, and how have their collaborative networks developed?

Most Relevant Sources

Figure 3 highlights the top relevant sources for tobacco-related content. The International Journal of Environmental Research and Public Health leads with 57 publications, followed by the Cochrane Database of Systematic Reviews (33) and PLOS ONE (31). Other notable sources include the International Journal of Molecular Sciences, Medicine, Addiction, and Nicotine & Tobacco Research. The presence of journals across public health, clinical, and molecular domains signifies the interdisciplinary scope of tobacco research, reflecting how health, behavioral science, and biomedical fields intersect in addressing tobacco use and its impacts. When compared with similar bibliometric reviews (e.g., Sharma et al., 2023; Li et al., 2020) [26,29], a consistent pattern emerges—IJERPH, PLOS ONE, and Addiction often appear among the top contributors, confirming their central role in tobacco-related dissemination. This also aligns with recent trends where public health journals are increasingly prioritizing tobacco control as a core issue.

Bradford’s Law

Figure 4 illustrates Bradford’s Law, which indicates that a small number of core journals produce the majority of tobacco-related publications. Introduced by Samuel Clement Bradford in the 1930s, this law ranks journal productivity by dividing them into zones, each containing an equal number of articles. The ratio between the core and outer zones follows the pattern 1:n:n^2^ […]^2^, helping analyze the distribution of literature across fields [36,37]. The top 10 journals in Zone 1 dominate the field, with the International Journal of Environmental Research and Public Health leading with 57 articles, followed by the Cochrane Database of Systematic Reviews with 33 articles and PLOS ONE with 31 articles. Other key contributors include Addiction, Nicotine & Tobacco Research, and BMC Public Health, highlighting the interdisciplinary nature of tobacco research across clinical, public health, and molecular sciences. This distribution underscores the central role of a few high-impact journals in shaping the field. This highlights the dominant role of a few critical journals in shaping research in the field.

Sources’ Production Over Time

Figure 5 shows that research on tobacco and public health has steadily grown in critical journals. It shows that by 2024, the International Journal of Environmental Research and Public Health had become the leading journal, surpassing its position after 2017. The International Journal of Molecular Sciences experienced a surge in post-2021, focusing on molecular research. PLOS ONE showed consistent growth, leveling off after 2018, while the Cochrane Database of Systematic Reviews consistently contributed through systematic reviews. These trends underscore increasing research in environmental, molecular, and review-based tobacco studies. These trends indicate a diversification in tobacco research, with increasing emphasis on environmental, molecular, and systematic review-based studies. The growth in publications highlights the evolving landscape of tobacco-related research, where interdisciplinary approaches are increasingly vital in understanding its health impacts. This trend is consistent with findings from other bibliometric studies (e.g., Xu et al., 2020) [38], which observed a rising convergence of environmental health and tobacco research, especially regarding secondhand smoke, e-cigarettes, and genetic effects. Numerous bibliometric analyses have similarly explored these intersections, reflecting a broader shift toward multidimensional perspectives in tobacco research. Recent studies have also identified emerging hotspots such as electronic health (eHealth), mobile health (mHealth), mental health, and the biological basis of nicotine addiction as key focal areas within smoking cessation research.

Author Contributions

Leading Contributors by Authorship

Figure 6 showcases the most prolific authors in tobacco research, with Wang Y and Zhang Y leading the field, each contributing 20 publications. Wang C (16) and Zhang L (15) also play significant roles, followed closely by Chen Y (14) and Wang X (13). Other notable contributors, including Liu Y (12), Wang Z (11), and Li J (10), further highlight the concentration of research output among a select group of highly active scholars. This trend underscores the pivotal role of a few key researchers in shaping tobacco-related studies, particularly in smoking and smokeless tobacco research. The dominance of these authors suggests strong academic networks, with significant contributions likely originating from leading research institutions in China and the U.S. These authors’ outputs are consistent with earlier bibliometric assessments (e.g., Yang et al., 2022) [39], which highlighted the dominance of Chinese researchers in both clinical and epidemiological tobacco studies. However, there is limited representation from African or Latin American scholars, which could signal a research gap or resource disparity in these regions.

Authors’ Production Over Time

Figure 7 shows the publication trends of individual authors in tobacco research over the years. Wang Y and Zhang Y have demonstrated consistent contributions to smoking and smokeless tobacco studies, maintaining a steady presence in the field. In contrast, Chen Y and Li J have contributed less frequently but have still made a significant impact. Chen Y’s research output has varied, with notable peaks in 2018 and 2024, while Li J has contributed sporadically in 2015 and 2018. The fluctuations in publication frequency suggest varying levels of engagement and influence over time, reflecting evolving research priorities and potential shifts in collaboration networks. As reported by Chen et al. (2020) [34], this kind of concentration can lead to intellectual silos, potentially narrowing perspectives. Efforts to decentralize research, provide funding to LMICs, and promote collaborative authorship may help diversify thought leadership in the field.

Lotka’s Law

Figure 8 illustrates the application of Lotka’s Law in tobacco research, showing that a small number of prolific authors contribute the majority of publications. In contrast, a large proportion of authors publish only one or two papers. This pattern aligns with the inverse square law, where the number of contributors decreases as their publication count increases. Specifically, 91.4% of authors have published only one paper, while a significantly smaller fraction (6.1%) has authored two. The number of highly productive authors drops sharply, with only a handful reaching six or more publications. This trend reinforces the concentrated nature of research productivity in the field, emphasizing the critical role of a small group of highly active researchers in advancing tobacco-related studies. The law highlights that a minority of highly active authors are responsible for most of the scientific output, reinforcing the concentrated nature of research productivity in tobacco-related studies [40]. As reported by Chen et al. (2020) [34], this kind of concentration can lead to intellectual silos, potentially narrowing perspectives. Efforts to decentralize research, provide funding to LMICs, and promote collaborative authorship may help diversify thought leadership in the field.

Most Relevant Affiliation

Figure 9 highlights the leading institutions in global tobacco research, with Harvard Medical School (114 publications) and the University of California, Los Angeles (104 publications) at the forefront. The United States dominates both single-country and collaborative studies, fostering significant international partnerships. The United Kingdom also makes notable contributions, with the University of Oxford (96 publications) playing a key role. Canada demonstrates strong research output through the University of Calgary (95) and the University of Toronto (81). Australia is represented by institutions such as the University of Newcastle (83) and Monash University (75), reflecting its active participation in tobacco-related studies. Meanwhile, China, Australia, and Italy contribute extensively but tend to focus more on single-country research with comparatively lower international collaborations. These leading institutions play a crucial role in shaping global tobacco control policies and public health strategies, reinforcing the significance of institutional contributions in addressing tobacco-related issues. The network visualization of keywords (Figure 10) displays thematic clusters and interconnections, with larger nodes indicating terms that are frequently used. This observation aligns with previous international research collaborations mapped by (33), which noted that high-income countries dominate tobacco research funding and output. In contrast, middle-income countries are underrepresented despite bearing a higher tobacco burden.

The figure represents a co-occurrence network of keywords, where node size indicates the frequency of each keyword and edge thickness represents the strength of co-occurrence. Different colors denote thematic clusters within the research field.

Publication Trajectory by Institutional Affiliation

Figure 11 shows that the Chan School of Public Health steadily increased its output to over 90 articles by 2023. Harvard Medical School’s growth was moderate, reaching 80 articles but slowing after 2021. Starting slowly, the University of Calgary accelerated its output post-2017 to get 90 articles. The University of California experienced early significant growth but plateaued after 2018, ending with 85 articles. The University of Oxford showed steady growth, with a notable rise after 2020, reaching 80 articles. All institutions demonstrated upward trends with varying growth rates. A comparison with the bibliometric review by Zhang et al. (2022) [25] reveals similar institutional trends but notes that publication rates from LMIC universities remain relatively stagnant. This again highlights the need for investment in global tobacco control research infrastructure.

RQ. 3. What are the most prevalent risk factors studied in smoking and smokeless tobacco research over the past 10 years, and how do these factors differ across geographical regions?

Over the last decade, research on smoking and smokeless tobacco has increasingly diversified in terms of the risk factors explored. Earlier in the decade, maternal and prenatal health risks dominated the focus, as indicated by recurring keywords such as “pregnancy”, “female”, and “risk factors”. These studies highlighted the adverse impacts of tobacco use during pregnancy, including low birth weight, miscarriage, and developmental disorders. However, a gradual shift in focus is observed in more recent years, with declining frequency of these keywords, suggesting a transition toward broader public health concerns and new demographic targets such as adolescents, young adults, and occupational exposure groups.

Corresponding Author’s Countries

Figure 12 shows that the United States leads global tobacco research with 450 publications, including 390 single-country publications (SCP), indicating strong domestic research efforts and international collaborations. China follows with 198 publications (169 SCP), while Australia (142 articles, 108 SCP) and Italy (115 articles, 85 SCP) also make significant contributions. Canada (84 articles, 67 SCP) and France (68 articles, 48 SCP) maintain a moderate presence. Iran (48 articles, 34 SCP), the Netherlands (47 articles, 32 SCP), the United Kingdom (47 articles, 29 SCP), and India (45 articles, 39 SCP) show lower but consistent contributions. While the U.S. dominates both single-country and collaborative research, other nations, particularly China, Australia, and Italy, focus more on domestic research.

Countries’ Production Over Time

Figure 13 illustrates the varying research outputs on tobacco studies from 2014 to 2023 across different countries. The fastest-growing country is expected to surpass 2000 articles by 2023, while another shows steady growth, reaching around 1500 articles. A third group demonstrates moderate growth, nearing 1000 articles, whereas others display a slower but steady increase, totaling around 500 articles. Some countries exhibit minimal growth, remaining below 500 articles. The United States leads in tobacco research, with a significant rise from 45 articles in 2014 to 2418 in 2024, followed by China, which increased from 27 articles to 1569 in the same period. Italy and Australia show moderate growth, reaching 940 and 900 articles, respectively, by 2024, while Canada records the slowest growth, with 609 articles. These data highlight the dominance of the U.S. and China in global tobacco research, while other nations contribute steadily at comparatively lower levels.

Countries’ Scientific Production

Figure 14 shows that the U.S. and China are the leading contributors to global research, followed by Canada, Australia, and Brazil, with significant involvement from Europe and India. Collaborative research is common among top-producing countries. The U.S. currently leads, but China is rapidly catching up and may match it by 2024. Australia, Canada, and Italy have also shown steady growth, with notable increases in publications post-2020, likely due to COVID-19-related research. Other countries, including India, Iran, and the Netherlands, maintain lower publication volumes, yet consistently address region-specific challenges such as gutkha use, cultural practices, and low-resource healthcare burdens. In summary, while maternal health once dominated tobacco research, the focus has broadened to include emerging risk groups and public health frameworks. The U.S. and China lead in both volume and thematic diversity, with region-specific priorities shaping the direction of research in other countries. The evolution of international collaboration and increased global output reflects an expanding and interconnected research community tackling tobacco-related harm from multiple angles.

RQ. 4. What are the emerging research frontiers and keyword trends in smoking and smokeless tobacco studies from 2014 to 2024, and how can these insights inform future research agendas and policy-making?

Most Frequent Words

The bibliometric analysis from 2014 to 2024 reveals evolving research trends and emerging frontiers in smoking and smokeless tobacco studies, offering critical insights for shaping future scientific inquiry and public health policy.

Figure 15 presents the bibliometric analysis, highlighting a strong focus on human-centered research, with the terms “humans” (2694 occurrences), “female” (2360), and “male” (1672) being the most frequent. Key themes include risk factors (875), pregnancy (652), smoking (252), and epidemiological studies (prevalence: 219, risk assessment: 162, prospective studies: 83). Age-related terms like “middle-aged” (582), “adolescent” (302), and “infant/newborn” (168) suggest a broad demographic scope. Smoking-related research is prominent, encompassing “smoking cessation” (92) and “electronic nicotine delivery systems” (73). Lifestyle and socioeconomic influences are evident in “diet” (70), “health behavior” (70), and “quality of life” (65). Genetic and environmental factors, such as genetic predisposition to disease (60) and environmental exposure (71), are also significant. The rise of COVID-19-related terms, such as “ACE2” and “SARS-CoV-2”, reflects recent health trends. Overall, the data underscores public health, disease prevention, and emerging concerns, including e-cigarettes and pandemic-related research.

Word Cloud

This word cloud visualizes the most frequently occurring keywords in the dataset, with larger words indicating higher frequency. Figure 16 highlights key research areas, with prominent terms like “smoking cessation”, “risk factors”, and “electronic cigarettes”. Central themes include “humans”, “female”, “male”, and “risk factors”, while terms like “pregnancy”, “middle-aged”, “adult”, and “adolescent” reflect a focus on various demographics. Additional terms such as “smoking”, “drug effects”, and “aged” emphasize public health, lifestyle risks, and aging. The word cloud underscores human health, gender, age, and risk factors. Smoking-related terms such as “smoking cessation”, “smoking/adverse effects”, and “electronic nicotine delivery systems” highlight the focus on tobacco use and cessation strategies. Lifestyle and environmental influences are evident through terms such as “diet”, “health behavior”, and “environmental exposure/adverse effects”. The presence of “comorbidity”, “genetic predisposition to disease”, and “socioeconomic factors” suggests an interest in multifactorial health determinants. The inclusion of “randomized controlled trials” and “cohort studies” highlights the methodological approaches employed in this research.

Tree Map

The tree map of the current review illustrates the distribution of key terms in the dataset, with larger blocks representing higher frequency. Figure 17’s tree map shows a strong focus on disease outcomes related to smoking and demographics. The most significant areas are “humans” (18%), “females” (16%), and “males” (11%). Other notable topics include “risk factors” (6%), “pregnancy” (4%), “adults” (5%), and “aged” (3%). Smaller topics like “smoking”, “exercise”, and “diet” are less emphasized. Rectangle sizes indicate the importance or prevalence of each topic.

Research Trends

Figure 18 highlights a growing focus on new tobacco products like e-cigarettes alongside traditional concerns such as lung cancer and cardiovascular diseases. Research trends evolved over time, with increasing attention to “electronic nicotine delivery systems”, “risk factors”, and “social determinants of health” after 2017. Topics like “odds ratio”, “aged 80 and over”, and “cross-sectional studies” peaked around 2017, while interest in “lung cancer” and “cardiovascular risk” declined after 2019. From 2021 to 2023, emerging topics included “surgical wound infections”, “quality of life”, and “feeding behavior”.

Conceptual Structure (Network Approach)

This conceptual structure network in Figure 19 visualizes the relationships between key terms in the dataset. Larger nodes represent frequently occurring terms such as “humans”, “female”, and “male”, while the connections indicate co-occurrence patterns, revealing thematic clusters in the research field. Figure 6’s co-occurrence network analysis reveals two main clusters: health impacts of tobacco (e.g., “lung cancer”, “cardiovascular disease”) and public health/prevention (e.g., “smoking cessation”, “public policy”). Key journals like the International Journal of Environmental Research and Public Health are experiencing significant growth. E-cigarettes have become a recent research hotspot. The Blue Cluster, led by researchers like Chen, Zhang, and Wang, is the largest collaboration group, while smaller clusters and isolated researchers contribute niche insights.

Thematic Map

Figure 20 thematic map shows high development but low relevance in the top-left quadrant, with topics such as “pregnancy”, “child”, and “animals”. The top-right quadrant is empty, indicating highly developed and relevant topics. The bottom-left quadrant is also empty, suggesting potential for emerging or declining themes. In the bottom-right quadrant, topics like “humans”, “female”, and “male” have high relevance but lower development. Central themes like “middle-aged”, “aged”, and “prevalence” are neutral or transitioning.

Social Network

Collaboration Network

Figure 21’s collaboration network reveals the U.S. as a central hub for international partnerships, with China and European countries also playing significant roles. India is increasing its global connections but remains less integrated. The co-authorship network shows strong collaboration, with red and blue clusters representing significant networks where larger nodes are key contributors. Isolated nodes indicate authors with fewer collaborations in niche areas. Thicker edges between nodes highlight strong partnerships and frequent co-authorship.

This network visualization illustrates the collaborative relationships among authors in the dataset. Nodes represent individual authors, while edges indicate co-authorship links. Different colors highlight distinct research clusters, reflecting collaborative groups working on related topics. The analysis reveals Cluster 1 as a dominant research group, with key contributors playing central roles in connecting researchers. Zhang Y (Betweenness: 34.472, PageRank: 0.035) and Chen Y (Betweenness: 29.327, PageRank: 0.032) emerge as significant network influencers, facilitating research connections across multiple teams. Wang H (Betweenness: 21.204, PageRank: 0.035), Guo Y, and Li H further strengthen the collaboration by acting as bridging nodes. While major nodes represent influential authors with frequent collaborations, smaller and isolated nodes indicate researchers with fewer co-authorships in specialized areas. The presence of strong co-authorship links (thicker edges) underscores well-established research partnerships, particularly within the red and blue clusters.

Countries’ Collaboration World Map

This map in Figure 22 visualizes international research collaborations, where nodes represent countries and connecting lines indicate co-authorship links between researchers from different nations. Darker shades highlight countries with higher research contributions, emphasizing strong global networks in the field. Figure 8 illustrates global collaboration patterns in tobacco research, with the U.S. and China at the center of extensive international networks. The U.S. leads in collaborative efforts, forging strong partnerships with European, Asian, and other nations, as evidenced by its numerous bilateral collaborations with countries such as Australia, Canada, and Italy. China also plays a central role, engaging in robust partnerships across continents. Emerging players, notably India, are becoming increasingly integrated into these networks, collaborating with the U.S., European countries, and other Asian nations. This global map of collaborations highlights both established and growing research relationships, underscoring the interconnected nature of contemporary tobacco research.

RQ. 5. How many publications and authors have contributed to research on the effects of smoking and smokeless tobacco in India over the past ten years, and what are the trends in authorship and collaboration?

India’s increasing contribution to tobacco-related research, projected to reach 255 publications by 2024, reflects the country’s growing recognition of the public health risks associated with tobacco use. Although India’s total number of publications still lags behind that of leading countries like the U.S. and China, the upward trend in research output, especially in areas related to smokeless tobacco and cancer, signals a shift in focus toward region-specific health challenges. This rise in publications is crucial as it highlights the emerging role of India in the global tobacco control landscape, particularly in understanding the unique socio-cultural and health issues related to tobacco consumption in the Indian context. India’s growing research output is also vital for shaping evidence-based tobacco policies tailored to the country’s needs, considering the significant prevalence of smokeless tobacco use, which is a leading cause of oral and digestive cancers. The increase in publications reflects both the increasing awareness of these health risks and the nation’s commitment to advancing tobacco control strategies. This shift suggests that India is not only contributing to the global body of research but is also taking a proactive role in addressing tobacco-related health disparities within its population. In a broader global context, India’s rise in tobacco research strengthens the international community’s efforts to tackle the global tobacco epidemic. It also provides valuable insights into the dynamics of tobacco consumption in low- and middle-income countries, which have unique challenges and patterns of use that differ from those in high-income nations.

## 4. Discussion

Over the past decade, research on smoking and smokeless tobacco has transformed significantly, driven by evolving public health priorities and advancements in tobacco-related products. Initially, the focus was on traditional health impacts such as lung cancer and cardiovascular diseases. However, the research landscape has broadened to include newer areas such as electronic nicotine delivery systems (e-cigarettes) and the social determinants of tobacco use. This shift reflects a deeper understanding of the risks associated with tobacco consumption, moving beyond direct health consequences to encompass social and technological factors. The surge in publications around 2021 likely corresponds to the COVID-19 pandemic, which highlighted the vulnerability of tobacco users to respiratory illnesses.

Bibliometric studies, when well conducted, offer valuable insights into the development of a research field, guiding future directions in meaningful and innovative ways. These studies enable researchers to gain a comprehensive overview of the literature, identify knowledge gaps, generate novel ideas for investigation, and position their contributions to the field. This study, through five key research questions, addresses these aspects to shed light on trends, gaps, and emerging opportunities in tobacco-related research:

1. One-Stop Overview: Evolution of Research Trends (RQ1)

Geographically, the focus of smoking and smokeless tobacco research varies. In the U.S. and Europe, lung cancer and cardiovascular diseases remain key concerns, whereas in Asia, smokeless tobacco research has increasingly centered on oral cancers and respiratory disorders. Notably, in India, smokeless tobacco use is linked to a rise in oral and digestive tract cancers, which are among the leading causes of cancer-related deaths in men [41]. Additionally, studies have noted that smoking adversely impacts reproductive health, with women who smoke during pregnancy facing a higher risk of complications and men experiencing increased risks of erectile dysfunction and sperm damage [42,43].

2. Identifying Leading Contributors and Collaborative Networks (RQ2)

Bibliometric analysis provides a clear overview of the key trends, emerging fields, and contributions of leading researchers and institutions in tobacco-related research. Countries such as the U.S., China, and various European nations have historically dominated tobacco research, producing the largest number of publications and fostering international collaborations. Prominent institutions like Harvard Medical School and the University of California have played crucial roles in steering the direction of this research, with influential authors such as Chen, Zhang, and Wang contributing extensively. Globally, tobacco consumption remains the leading contributor to preventable diseases and deaths, reinforcing the need for ongoing and coordinated research efforts [44].

3. Identifying Knowledge Gaps and Emerging Research Frontiers (RQ3 and RQ4)

Keyword analysis reveals a rising focus on public health responses to novel tobacco products [45,46]. However, the widespread use of e-cigarettes among young people, especially in Europe, coupled with the misconception that they are risk-free, underscores the need for stronger public health campaigns and policy responses. This emerging concern suggests that future research will likely focus on policy-making related to the regulation of new tobacco products, particularly among younger populations.

4. Deriving Novel Ideas for Investigation: Health Implications of E-Cigarettes (RQ3)

Future studies should focus on the long-term health consequences of e-cigarette use, particularly among adolescents and young adults. Furthermore, research exploring strategies such as smoking cessation programs targeting e-cigarette users could provide valuable insights into controlling tobacco-related risks.

5. Positioning Contributions to the Field: India’s Role in Tobacco Research (RQ5)

India’s increasing contribution to global tobacco research, projected to reach 255 publications by 2024, highlights the country’s growing recognition of tobacco-related health issues. While India’s research output is still smaller compared with leading countries, the rise in publications, particularly those focused on smokeless tobacco and cancer outcomes, signals India’s emerging role in global tobacco control strategies. Studies on the environmental and societal impacts of tobacco use further emphasize the urgent need for a multisectoral approach to tobacco control in India. Tobacco use in India not only takes lives but also negatively affects the environment, society, and economy [47].

This bibliometric review contributes to the field by synthesizing a decade of global and Indian research on tobacco use, highlighting the need for further investigation into e-cigarettes and smokeless tobacco. It positions future research within the context of emerging trends and health concerns, particularly in light of the increasing popularity of e-cigarettes. Given the growing use of e-cigarettes, especially in Europe, addressing the misconception among young people that they are harmless is a critical area for policy development. Future research should also focus on the regulation of emerging tobacco products and their impact on public health, particularly in vulnerable populations such as youth and pregnant women.

In summary, the evolution of tobacco research from 2014 to 2024 reveals a dynamic and expanding field. New health concerns, technological innovations, and shifting public health priorities have influenced research directions. The increase in global collaboration, regional focus on specific health risks, and the emergence of new research areas, such as e-cigarettes, point toward a comprehensive approach to mitigating tobacco-related harms in the coming years. These findings are crucial for shaping future research agendas and developing effective tobacco control policies worldwide.

## 5. Conclusions

In conclusion, the bibliometric analysis of smoking and smoke tobacco research from 2014 to 2024 reveals a significant growth in scientific output, reflecting increasing global attention to tobacco-related health issues. The shift in research focus from traditional tobacco harms, such as lung cancer and cardiovascular diseases, to newer concerns like electronic nicotine delivery systems highlights the evolving landscape of public health challenges. Leading institutions and countries, particularly the U.S. and China, continue to drive progress, while India’s growing contribution marks its rising importance in global tobacco research. These insights underscore the need for continued collaboration and innovation to address the complex health impacts of tobacco use, with future research likely to focus on emerging products, social determinants, and broader public health implications.

### 5.1. Clinical Implications

The expanding research on tobacco products, including e-cigarettes and smokeless tobacco, emphasizes the need for healthcare professionals to stay updated on emerging risks and treatment strategies. The increasing volume of research underscores the importance of integrating findings into clinical practice, particularly in the areas of smoking cessation interventions. Clinicians should consider the regional variations in tobacco-related health impacts and tailor preventive and therapeutic approaches accordingly. Enhanced collaboration and continued research are crucial to addressing the evolving challenges in tobacco control and improving patient outcomes.

### 5.2. Limitations and Benefits

This study has its limitations. Primarily, it relies solely on PubMed, which may overlook important studies from other relevant databases. Additionally, the absence of citation metrics limits the assessment of research quality and impact, focusing only on publication trends. The use of “Keywords Plus” might also miss certain specific research trends. Furthermore, the predominance of research conducted in the U.S. and China could skew the findings, marginalizing contributions from smaller nations. Lastly, the evolving terminology associated with tobacco products may impact the accuracy of trend analysis. Future studies should incorporate multiple databases and citation metrics for a more comprehensive analysis. Including non-English publications would also help reduce language bias and encompass significant studies in other languages. Despite these limitations, this study provides a valuable overview of trends in smoking and smokeless tobacco research, identifies emerging issues, and offers insights into regional public health strategies and future research priorities.

## Figures and Tables

**Figure 1 healthcare-13-01224-f001:**
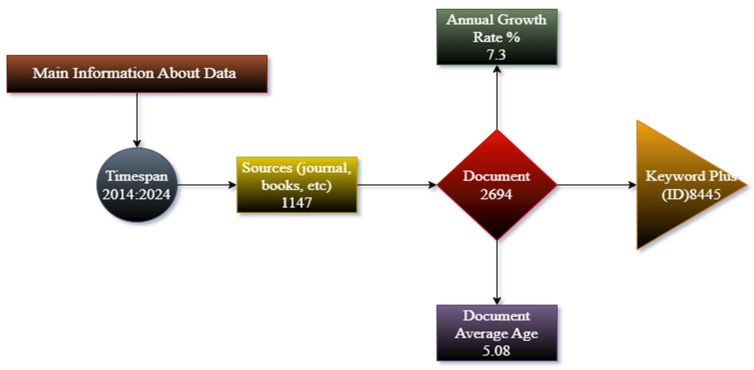
Overview of key data attributes.

**Figure 2 healthcare-13-01224-f002:**
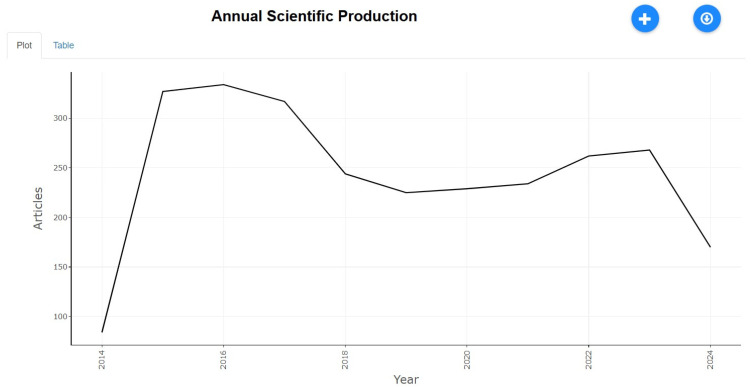
Annual scientific production.

**Figure 3 healthcare-13-01224-f003:**
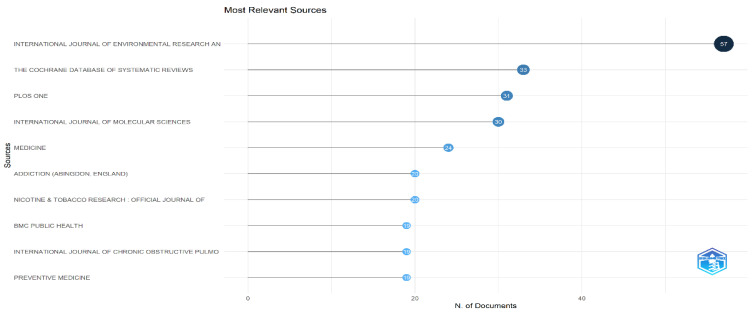
Most relevant sources for published documents.

**Figure 4 healthcare-13-01224-f004:**
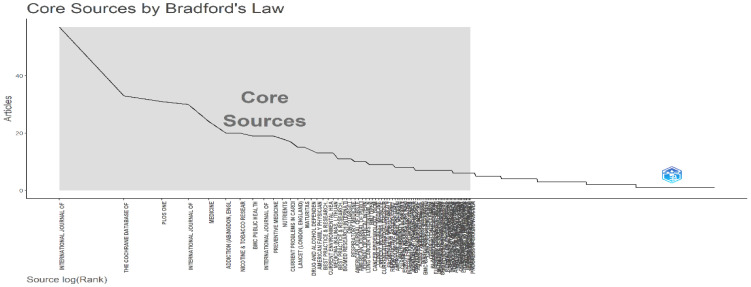
Sources by Bradford’s Law.

**Figure 5 healthcare-13-01224-f005:**
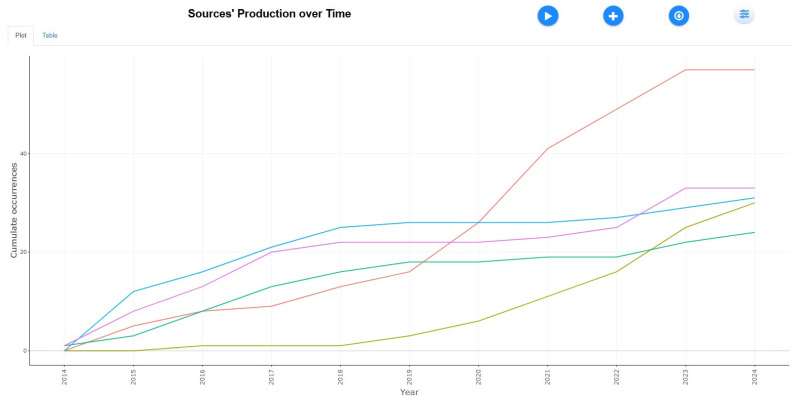
Sources’ production over time.

**Figure 6 healthcare-13-01224-f006:**
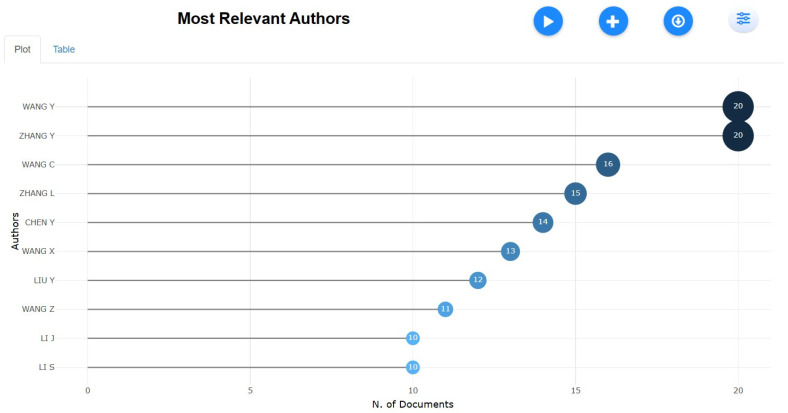
Leading contributors by authorship.

**Figure 7 healthcare-13-01224-f007:**
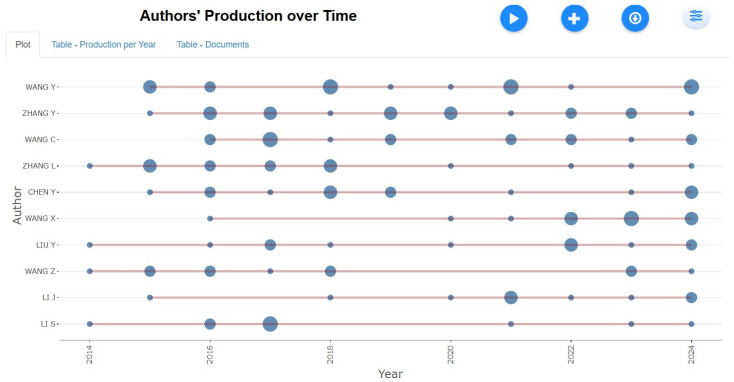
Authors’ production over time.

**Figure 8 healthcare-13-01224-f008:**
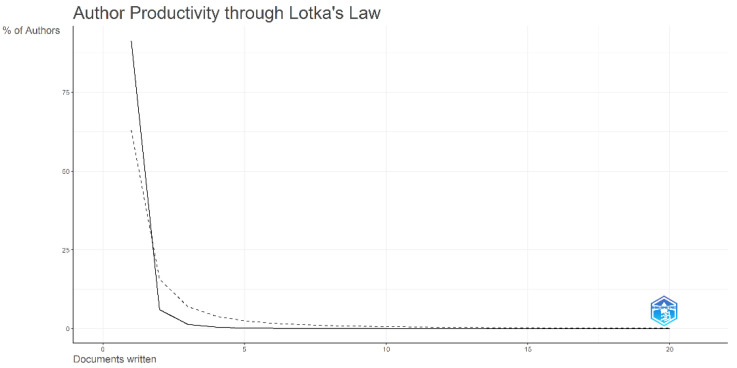
Authors’ productivity through Lotka’s Law.

**Figure 9 healthcare-13-01224-f009:**
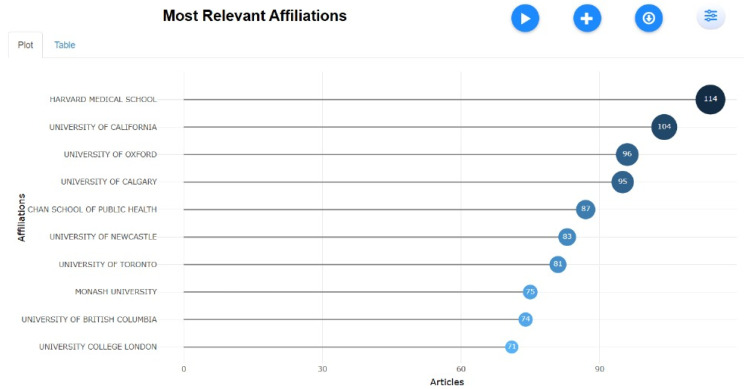
Most relevant affiliations.

**Figure 10 healthcare-13-01224-f010:**
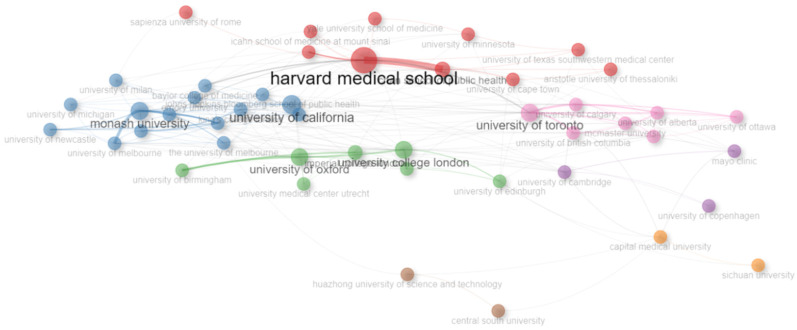
Most relevant affiliations network.

**Figure 11 healthcare-13-01224-f011:**
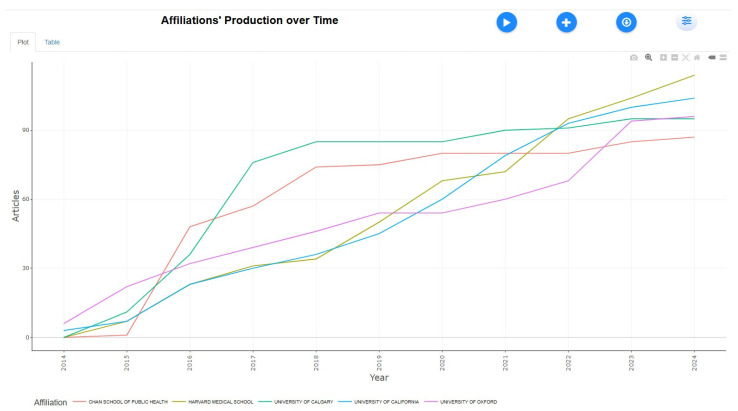
Affiliations’ production over time.

**Figure 12 healthcare-13-01224-f012:**
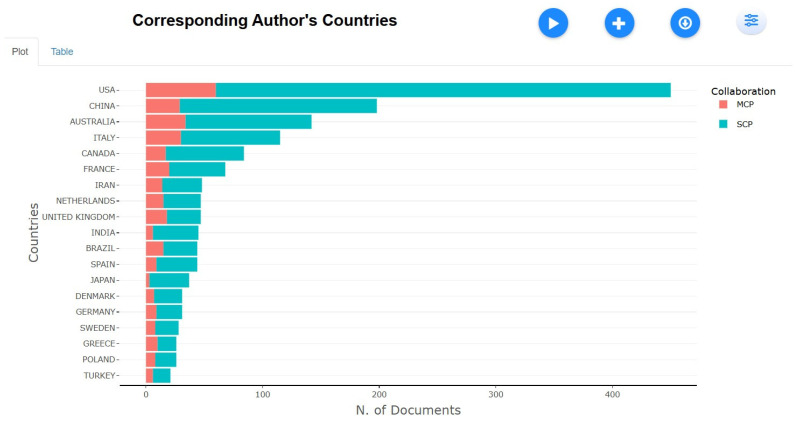
Corresponding authors’ countries.

**Figure 13 healthcare-13-01224-f013:**
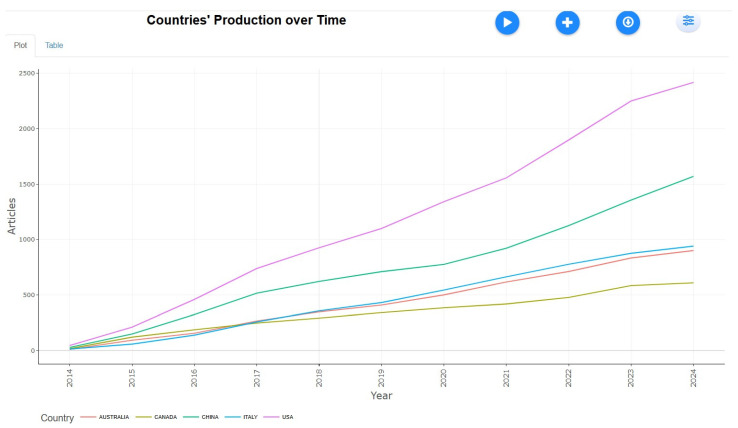
Countries’ production over time.

**Figure 14 healthcare-13-01224-f014:**
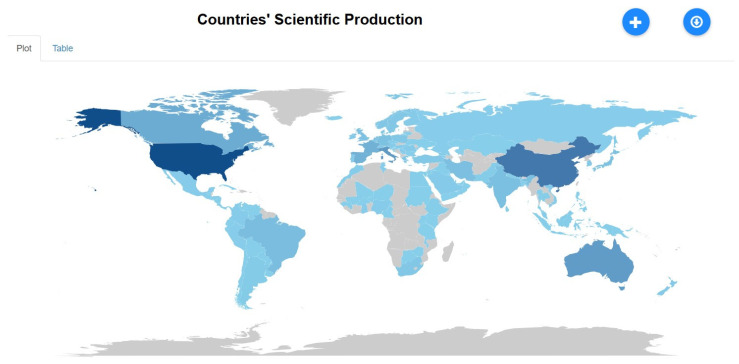
Global scientific production by country.

**Figure 15 healthcare-13-01224-f015:**
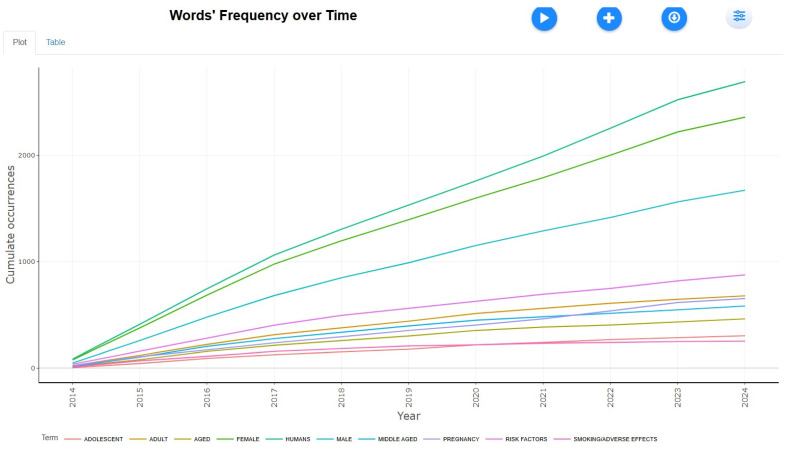
Most frequent words.

**Figure 16 healthcare-13-01224-f016:**
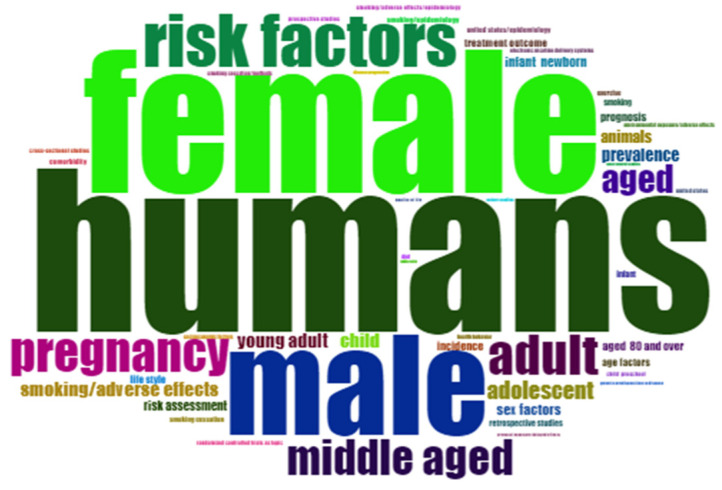
Word cloud representation of key terms.

**Figure 17 healthcare-13-01224-f017:**
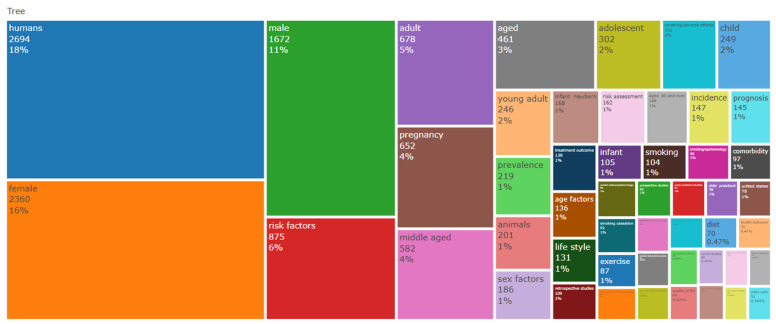
Tree map representation of key terms.

**Figure 18 healthcare-13-01224-f018:**
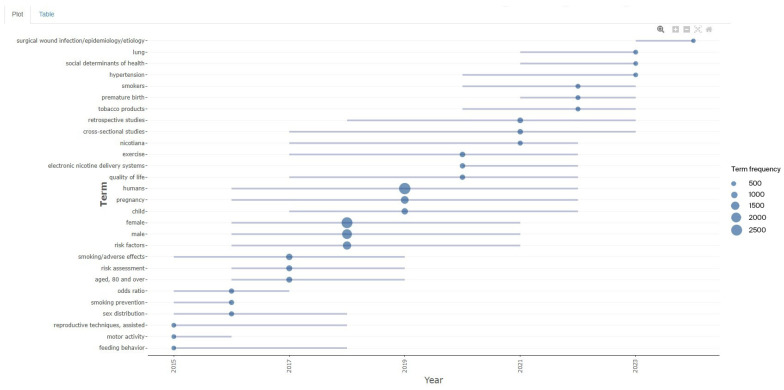
Research trends.

**Figure 19 healthcare-13-01224-f019:**
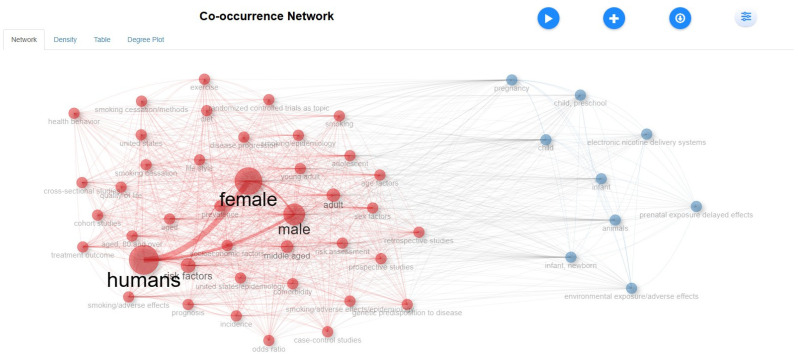
Conceptual structure (network approach).

**Figure 20 healthcare-13-01224-f020:**
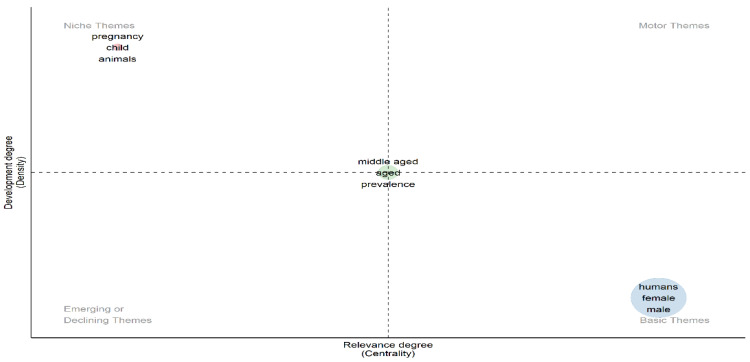
Thematic map.

**Figure 21 healthcare-13-01224-f021:**
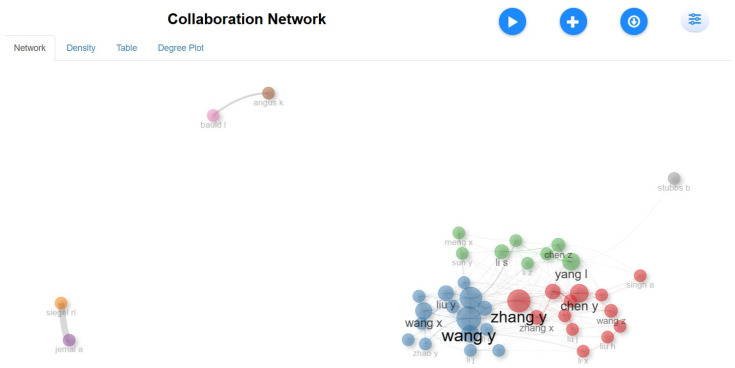
Authors’ collaboration network.

**Figure 22 healthcare-13-01224-f022:**
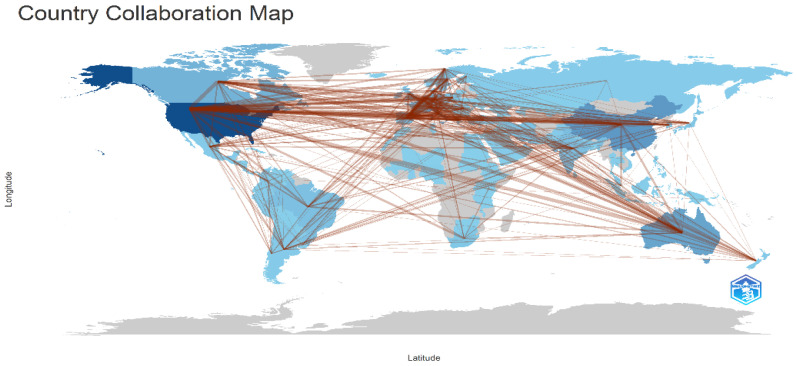
Countries’ collaboration world map.

**Table 1 healthcare-13-01224-t001:** Research questions with their significance.

Research Questions	Significance
How have research trends and focus areas in smoking and smokeless tobacco studies evolved over the past 10 years (2014–2024)?	This question examines how research topics have developed over the last decade, identifying key trends and shifts in focus within smoking and smokeless tobacco studies.
2. Which authors, institutions, and countries have been most influential in smoking and smokeless tobacco research over the last 10 years, and how have their collaborative networks developed?	This question explores the field’s leading contributors and collaboration patterns over the past decade, focusing on geographical and institutional collaborations.
3. What are the most prevalent risk factors studied in smoking and smokeless tobacco research over the past 10 years, and how do these factors differ across geographical regions?	This question compares tobacco research’s most frequently studied risk factors, focusing on regional variations over the last decade.
4. What are the emerging research frontiers and keyword trends in smoking and smokeless tobacco studies from 2014 to 2024, and how can these insights inform future research agendas and policy-making?	This question uses keyword analysis to identify emerging topics and research priorities, offering insights into future research directions and policy development.
5. How many publications and authors have contributed to research on the effects of smoking and smokeless tobacco in India over the past ten years, and what are the trends in authorship and collaboration?	This question is significant as it highlights the research volume, key contributors, and collaboration trends in India’s tobacco research, guiding future efforts to address tobacco-related health issues.

Source: authors’ calculation.

## Data Availability

Data sharing is not applicable. No new data were created or analyzed in this study.

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
