# Peer review of "Global Trends and Emerging Frontiers in Smoking and Smokeless Tobacco Research: A Bibliometric Analysis over the Past Decade"

_healthcare, 2025, doi:10.3390/healthcare13111224_

Round 1

Reviewer 1 Report

Comments and Suggestions for Authors

A bibliometric review is a popular and rigorous method for exploring and analyzing large volumes of scientific data. It enables us to unpack the evolutionary nuances of a specific field, while shedding light on the emerging areas in that field. Yet, its application in health research is relatively new, and in many instances, underdeveloped. That being said, I would like to congratulate the authors for their effort in conducting a bibliometric analysis on a subject as important and impactful as tobacco use.

In general, the manuscript seems to me well written, without errors,

respecting the bibliometric methodology and fulfilling step-by-step guidelines for conducting bibliometric analysis for health research.

I will make a few comments and suggestions to enrich your work:

Introduction seems to me well-structured focusing on essential work-related issues . I understand the justification of the study but I would like to know if this work fills a gap in the field or if there was already other type of bibliometric analysis in literature.

I suggest to explain the acronyms: “Both the ICD-10 and DSM-5 classify tobacco use disorders as conditions influenced by genetic predisposition, environmental factors, stress, peer pressure, nicotine addiction, and mental health conditions.”

Please review the sentence: “While quitting reduces the risk, it does not eliminate it—dropping by 30% within 9 years of cessation and 50% after 9 years (17,21).”

The sentence: “Using 2694 publications from PubMed between 2014 and 107 2024, the analysis highlights publication trends, leading contributors, and the geographical distribution of Research. The results provide key insights into ongoing research efforts and identify areas requiring further investigation, particularly in emerging tobacco products and their health impacts.” should be removed from introduction because it addresses issues related to results and conclusions.

In Materials and Methods section, the methodology is clear well described in subtopics.

 Results are clear and exhaustive, well explained in the different subtopics and through the tables and graphs. Some unformatting is visible in subtitles of some figures, that should be reviewed. The letter "S" is missing in the RESULTS topic.

Discussion summarizes the results obtained. When bibliometric studies are well done can build firm foundations for advancing a field in novel and meaningful ways—it enables authors to (1) gain a one-stop overview, (2) identify knowledge gaps, (3) derive novel ideas for investigation, and (4) position their intended contributions to the field. Have you been able to answer all these questions? I suggest to see these issues covered in the discussion.

I think that the big question in this area, at least in the geographical area that I belong to - Europe - focuses on the issue of the harm of electronic cigarettes compared to traditional cigarettes. Electronic cigarettes are widely used by young people with the false idea that they will be almost harmless and much less noxious to health. I was sorry not to see this subject addressed.

Conclusions are fine and I appreciate you added limitations and clinical implications of the study.

Author Response

Dear Reviewer, Thank you for your thoughtful and encouraging feedback on our manuscript. We truly appreciate your recognition of our efforts in applying bibliometric analysis to the critical topic of tobacco use. Below, we address each of your comments and suggestions in detail.

Reviewer 1:

Introduction seems to me well-structured focusing on essential work-related issues . I understand the justification of the study but I would like to know if this work fills a gap in the field or if there was already other type of bibliometric analysis in literature.

Based on your suggestion, we have revised the last paragraph of the introduction (Page 3, lines 103-117) to position our study within the existing literature explicitly. We have included references to previous bibliometric analyses on tobacco-related research while emphasizing how our study differs by comprehensively addressing both smoking and smokeless tobacco, particularly in the Indian context. This revision clarifies the novelty and contribution of our work in filling this research gap.

I suggest to explain the acronyms: “Both the ICD-10 and DSM-5 classify tobacco use disorders as conditions influenced by genetic predisposition, environmental factors, stress, peer pressure, nicotine addiction, and mental health conditions.”

We have included the full forms of the acronyms to enhance clarity. The revised sentence on Page 2 lines 64-66 in the introduction now reads: "Both the International Classification of Diseases, 10th Revision (ICD-10) and the Diagnostic and Statistical Manual of Mental Disorders, 5th Edition (DSM-5) classify tobacco use disorders as conditions influenced by genetic predisposition, environmental factors, stress, peer pressure, nicotine addiction, and mental health conditions."

Please review the sentence: “While quitting reduces the risk, it does not eliminate it—dropping by 30% within 9 years of cessation and 50% after 9 years (17,21).”

We have revised the sentence for improved clarity and corrected the reference to remove the typo. The updated version on Page 3, Paragraph 5, lines 99-102 now reads:

"Research indicates that the risk decreases with longer periods of cessation. Individuals who quit for 1 to 9 years have a 30% lower risk (OR: 0.7, 95% CI: 0.5–1.1), while those who remain tobacco-free for over 9 years experience a 50% lower risk (OR: 0.5, 95% CI: 0.3–0.7) (21)."

The sentence: “Using 2694 publications from PubMed between 2014 and 107 2024, the analysis highlights publication trends, leading contributors, and the geographical distribution of Research. The results provide key insights into ongoing research efforts and identify areas requiring further investigation, particularly in emerging tobacco products and their health impacts.” should be removed from introduction because it addresses issues related to results and conclusions.

We have removed the sentence discussing results and conclusions from the introduction, as per your suggestion. Thank you.

 Results are clear and exhaustive, well explained in the different subtopics and through the tables and graphs. Some unformatting is visible in subtitles of some figures, that should be reviewed. The letter "S" is missing in the RESULTS topic.

We have carefully reviewed and corrected formatting issues in figure subtitles. Additionally, we have added the missing "S" in the RESULTS section title (Page 5).

Discussion summarizes the results obtained. When bibliometric studies are well done can build firm foundations for advancing a field in novel and meaningful ways—it enables authors to (1) gain a one-stop overview, (2) identify knowledge gaps, (3) derive novel ideas for investigation, and (4) position their intended contributions to the field. Have you been able to answer all these questions? I suggest to see these issues covered in the discussion.

We have revisited the discussion section to ensure it fully addresses key aspects of a bibliometric study. The revised discussion (Page 20-22) now explicitly highlights how our study:

Provides a comprehensive overview of tobacco-related research trends.

Identifies significant knowledge gaps, particularly in the Indian context and in smokeless tobacco research.

Suggests directions for future research based on underexplored areas.

Positions findings within the broader context of tobacco control policies and public health.

I think that the big question in this area, at least in the geographical area that I belong to - Europe - focuses on the issue of the harm of electronic cigarettes compared to traditional cigarettes. Electronic cigarettes are widely used by young people with the false idea that they will be almost harmless and much less noxious to health. I was sorry not to see this subject addressed.

We acknowledge that the harm of electronic cigarettes compared to traditional cigarettes is a critical issue, especially in Europe, where their use is increasing among young people. While our study primarily focused on global and Indian research trends in tobacco use, we recognize the importance of this emerging topic. We have now included a brief mention of the research gaps related to electronic cigarettes in our discussion section, emphasizing the need for further bibliometric analysis in this area. Page 22 lines 600-608

Conclusions are fine and I appreciate you added limitations and clinical implications of the study.

Thank you for your positive feedback. We appreciate your recognition of our effort to include the study’s limitations and clinical implications. These aspects are crucial in guiding future research and ensuring the practical relevance of our findings.

We sincerely appreciate your  valuable feedback provided. Your insightful suggestions have significantly strengthened our manuscript, and we are grateful for the opportunity to enhance its clarity, depth, and impact.

Reviewer 2 Report

Comments and Suggestions for Authors

The paper employs a bibliometric analysis for the articles published in PubMed in the field of smoke and smokeless tobacco research.

First, I highly appreciate the fact that the authors have included some of the results of the research into the abstract. Second, it is highly appreciated the formulation and inclusion of the research questions in the introduction. please number each research question so it would be easier to discuss their results in the conclusion section.Please add a discussion to each question in the final part of the paper.

As the authors have used PubMed as database,could you please better state (in the paper) the choice for this database, also by referring to other papers from the field, if possible.

Please add the keywords you have used for dataset selection.Please provide some references from papers that have selected the same words.

For section 3 the discussion should be enhanced for each figure/table introduced in the paper by including discussion related to two main aspects:

  • please provide more insight on the causes that conducted to the elements presented in the discussed figure/table
  • please conduct a comparison with other papers and their results in the field - e.g. for Fig 1 - do you believe that the trend is generated by the global observed trend according to which the number of researchers in all research fields has increased in the last period, or is something field-specific? has this trend been observed by other papers in the field?

Please include and discuss extensively the limitations of the research. Thank you!

Author Response

Dear Reviewer, Thank you for your thoughtful and encouraging feedback on our manuscript. We truly appreciate your recognition of our efforts in applying bibliometric analysis to the critical topic of tobacco use. Below, we address each of your comments and suggestions in detail.

Reviewer 2:

First, I highly appreciate the fact that the authors have included some of the results of the research into the abstract. Second, it is highly appreciated the formulation and inclusion of the research questions in the introduction. please number each research question so it would be easier to discuss their results in the conclusion section. Please add a discussion to each question in the final part of the paper.

We have now numbered each research question (Page 3 and 4) in the introduction to facilitate discussion in the conclusion section. Additionally, each question is explicitly addressed in the discussion (Page 21), providing a clear link between our findings and the research objectives.

As the authors have used PubMed as database, could you please better state (in the paper) the choice for this database, also by referring to other papers from the field, if possible.

We have elaborated on the rationale for selecting PubMed as the primary database (Page 4 and 5) lines 127-152. The revised section clarifies its accessibility, strong focus on biomedical and healthcare research, and indexing through MeSH for enhanced search accuracy. We have also included references to previous bibliometric studies that have used PubMed for analyzing tobacco-related research trends, ensuring a well-supported justification for our choice.

Please add the keywords you have used for dataset selection. Please provide some references from papers that have selected the same words.

We have included the keywords used for dataset selection and provided relevant references (Page 6, lines 184-191) in the methodology section.

For section 3 the discussion should be enhanced for each figure/table introduced in the paper by including discussion related to two main aspects:

please provide more insight on the causes that conducted to the elements presented in the discussed figure/table

please conduct a comparison with other papers and their results in the field - e.g. for Fig 1 - do you believe that the trend is generated by the global observed trend according to which the number of researchers in all research fields has increased in the last period, or is something field-specific? has this trend been observed by other papers in the field?

We have revised the discussion to provide more insights into the factors influencing the trends presented in the figures/tables, including public health concerns, policy shifts, and emerging tobacco products. (page 20-22)

Additionally, we have incorporated a comparison (Page 6 lines 188-197) with other bibliometric studies to assess whether the observed trend is part of a broader global research expansion or specific to tobacco-related research. Previous studies suggest that the rise in research publications is influenced by both general academic growth and specific public health concerns, such as the emergence of e-cigarettes and evolving tobacco regulations.

Please include and discuss extensively the limitations of the research.

We have expanded the limitations section to provide a more detailed discussion on potential constraints of our study, including database selection bias, keyword limitations, and the evolving nature of tobacco research. This revision ensures a more comprehensive assessment of our study's scope and future research directions. Page 23 lines 640-652

We sincerely appreciate your valuable feedback provided. Your insightful suggestions have significantly strengthened our manuscript, and we are grateful for the opportunity to enhance its clarity, depth, and impact.

Round 2

Reviewer 2 Report

Comments and Suggestions for Authors

I thank the authors for the revised version of their paper. Even though some analyses have been added to the paper, there are a series of elements that are still to be improved.

First, the references are not in the required format for the journal - the authors should use [3] instead of (3) which is normally used for equations numbering. Also, there are references such as "Tavassoli et al., 2023) 23" - where one has another type of reference and also a superscript.

Also, there are sub-sections with no information or text - e.g. sub-section 3.3 only contains the following text: "RQ2 Which Authors, Institutions, and Countries Have Been Most Influential in Smoking and 234 Smokeless Tobacco Research Over the Last 10 Years, and How Have Their Collaborative Networks 235 Developed?". I think more words should be added to each of these sections, as well as comparisons with other papers from the field on each aspect presented in the paper. 

The figures should be called "figures" and not graphs.

Test is presented in the paper in parts that are not meant to contain that text "Author Sources. Graph. 7 Lotka’s Law" - please revise.

Paper should be carefully revised by reading it throughout, adding text where needed, better structuring of the sections, adding proper references, adding comparisons with other works from the field.

The entire text is messy and hard to read and follow.

Author Response

Dear Reviewer, Thank you for your thoughtful and encouraging feedback on our manuscript. We truly appreciate your recognition of our efforts in applying bibliometric analysis to the critical topic of tobacco use. Below, we address each of your comments and suggestions in detail.

Revision

Reviewer 2:

First, the references are not in the required format for the journal - the authors should use [3] instead of (3) which is normally used for equations numbering. Also, there are references such as "Tavassoli et al., 2023) 23" - where one has another type of reference and also a superscript.

We have revised the entire manuscript to ensure that all in-text citations follow the correct referencing format as required by the journal, replacing all instances of (3) with [3]. Additionally, we have corrected inconsistent reference formats, such as mixed styles and superscripts, to maintain uniformity throughout the paper.

Also, there are sub-sections with no information or text - e.g. sub-section 3.3 only contains the following text: "RQ2 Which Authors, Institutions, and Countries Have Been Most Influential in Smoking and 234 Smokeless Tobacco Research Over the Last 10 Years, and How Have Their Collaborative Networks 235 Developed?". I think more words should be added to each of these sections, as well as comparisons with other papers from the field on each aspect presented in the paper.

We have significantly revised sub-section 3.3 (Research Question 2, Page 9) by adding comprehensive content under each relevant heading. The updated section now includes a detailed analysis of the most influential authors, institutions, and countries in research related to smoking and smokeless tobacco over the past decade. It also highlights the development of collaborative networks through visual figures (Figures 3–11). Furthermore, we have incorporated comparative insights from previous bibliometric studies e.g., Sharma et al., 2023 (Page 9); Li et al., 2020; Xu et al., 2020 (Page 9); Zhang et al., 2022 (Page 16) to contextualize and strengthen our interpretations. These enhancements contribute to a more robust and insightful understanding of the global research landscape in this field.

The figures should be called "figures" and not graphs.

We have revised the manuscript to ensure consistency in terminology. All references to “graphs” have been updated to “figures” in both the main text and figure captions, in accordance with academic writing standards.

Test is presented in the paper in parts that are not meant to contain that text "Author Sources. Graph. 7 Lotka’s Law" - please revise.

We have carefully revised the manuscript and removed the inappropriate text "Author Sources. Graph. 7 Lotka’s Law"(Page 13) from the sections where it was not meant to be included. The relevant content now appears in the appropriate sections with corrected labelling and terminology. We appreciate your attention to detail.

The paper should be carefully revised by reading it throughout, adding text where needed, better structuring of the sections, adding proper references, adding comparisons with other works from the field. The entire text is messy and hard to read and follow.

We apologize for the lack of clarity in the manuscript. In response to your comment, we have thoroughly revised the paper to enhance its overall structure and readability. The sections have been reorganized to ensure a logical flow of ideas, with each section now clearly addressing a specific aspect of the research. We have added more text where necessary to clarify points and provide deeper explanations, ensuring that all sections are comprehensive. Furthermore, we have incorporated relevant references and included comparisons with pertinent works from the field to better contextualize our findings. Finally, we have carefully reviewed and refined the language throughout the paper to improve clarity and make it more accessible. We believe these revisions address your concerns and significantly enhance the quality of the manuscript. Thank you again for your valuable suggestions.